

# Characterization of dengue cases among patients with an acute illness, Central Department, Paraguay

Alejandra Rojas[1], Fátima Cardozo[2], César Cantero[1], Victoria Stittleburg[3], Sanny López[1], Cynthia Bernal[1], Francisco Eugenio Gimenez Acosta[4], Laura Mendoza[2], Benjamin A. Pinsky[5,6], Ivalena Arévalo de Guillén[1], Malvina Páez[2] and Jesse Waggoner[3,7]

[1] Departamento de Producción, Instituto de Investigaciones en Ciencias de la Salud, Universidad Nacional de Asunción, San Lorenzo, Paraguay
[2] Departamento de Salud Pública, Instituto de Investigaciones en Ciencias de la Salud, Universidad Nacional de Asunción, San Lorenzo, Paraguay
[3] Department of Medicine, Division of Infectious Diseases, Emory University, Atlanta, GA, United States of America
[4] Área Ambulatoria, Hospital Distrial de Villa Elisa, Asunción, Paraguay
[5] Department of Pathology, Stanford University School of Medicine, Stanford, CA, United States of America
[6] Department of Medicine, Division of Infectious Diseases and Geographic Medicine, Stanford University School of Medicine, Stanford, CA, United States of America
[7] Rollins School of Public Health, Department of Global Health, Emory University, Atlanta, GA, United States of America

Corresponding authors
Alejandra Rojas,
alerojaspy@gmail.com
Jesse Waggoner,
jesse.waggoner@emoryhealthcare.org

## ABSTRACT

**Background.** In 2018, Paraguay experienced a large dengue virus (DENV) outbreak. The primary objective of this study was to characterize dengue cases in the Central Department, where the majority of cases occur, and identify factors associated with DENV infection.

**Methods.** Patients were enrolled from January-May 2018 if they presented with a suspected arboviral illness. Acute-phase specimens ($\leq 8$ days after symptom onset) were tested using rRT-PCR, a rapid diagnostic test for DENV nonstructural protein 1 (NS1) and anti-DENV IgM and IgG, and ELISA for IgG against NS1 from Zika virus (ZIKV).

**Results.** A total of 231 patients were enrolled (95.2% adults) at two sites: emergency care and an outpatient clinical site. Patients included 119 (51.5%) dengue cases confirmed by rRT-PCR ($n = 115$, 96.6%) and/or the detection of NS1 and anti-DENV IgM ($n = 4$, 3.4%). DENV-1 was the predominant serotype (109/115, 94.8%). Epidemiologically, dengue cases and non-dengue cases were similar, though dengue cases were less likely to reside in a house/apartment or report a previous dengue case. Clinical and laboratory findings associated with dengue included red eyes, absence of sore throat, leucopenia and thrombocytopenia. At an emergency care site, 26% of dengue cases (26/100) required hospitalization. In univariate analysis, hospitalization was associated with increased viral load, anti-DENV IgG, and thrombocytopenia. Among dengue cases that tested positive for IgG against ZIKV NS1, the odds of DENV NS1 detection in the acute phase were decreased 10-fold (OR 0.1, 0.0–0.3).

**Conclusions.** Findings from a predominantly adult population demonstrate clinical and laboratory factors associated with DENV infections and the potential severity of

dengue in this group. The combination of viral load and specific IgG antibodies warrant further study as a prognostic to identify patients at risk for severe disease.

## INTRODUCTION

Dengue is the commonest human arboviral disease worldwide, with an estimated 50–100 million cases occurring annually throughout the tropics and subtropics (*Stanaway et al., 2016*; *World Health Organization, 2009*). Dengue results from human infection with one of four related serotypes of dengue virus (DENV-1-4) (*Guzman & Harris, 2015*). In the five years leading up to and including the current study (2018), all four serotypes circulated in the region of South America surrounding Paraguay (Fig. 1), which reports among the highest annual incidence rates of dengue on the continent (*Dantes, Farfan-Ale & Sarti, 2014*; *Gordon et al., 2013*;*Pan American Health Organization, 2018*). Over the past decade, DENV-1 has circulated in Paraguay in all but one year, and it has been predominant since 2015 (*Dirección General de Vigilancia de la Salud, and Ministerio de Salud Pública y Bienestar Social, 2017*; *Pan American Health Organization, 2018*). Despite significant declines in dengue incidence throughout the Americas following the 2015–2016 Zika virus (ZIKV) epidemic, Paraguay experienced large numbers of dengue cases in 2016 and again in 2018 (*Pan American Health Organization, 2018*; *Perez et al., 2019*). These data suggest that arboviral epidemiology may be relatively unique in Paraguay, which is located at the southern boundary of the DENV-endemic region in the Americas (*Bhatt et al., 2013*; *Stanaway et al., 2016*; *World Health Organization, 2009*). However, relatively little data has been published on dengue in the country, and the majority of available data has come either from hospitalized pediatric cases or from international studies with only a subset of patients from Paraguay (*Halsey et al., 2012*; *Lovera et al., 2014*; *Lovera et al., 2016*; *Rojas et al., 2016*).

Symptomatic DENV infections classically present as an acute fever with myalgias and rash (*Guzman & Harris, 2015*; *World Health Organization, 2009*). However, patients can develop a wide array of signs and symptoms, which limits the accuracy of a clinical diagnosis based on exam findings and results of routine laboratory testing (*Gregory et al., 2010*; *Morch et al., 2017*; *Potts & Rothman, 2008*; *Waggoner et al., 2016b*). In addition, dengue manifests differently among children and adults, and factors associated with dengue cases and severe disease in a pediatric population may not be applicable in older patients (*Gregory et al., 2010*; *Hammond et al., 2005*; *Kittigul et al., 2007*). The differential diagnosis for dengue includes arboviral pathogens, such as chikungunya virus (CHIKV) and Zika virus (ZIKV), and local endemic diseases, such as leptospirosis, which may all cause an indistinguishable clinical picture (*Silva et al., 2018*; *Waggoner et al., 2016b*). Accurate diagnosis in the acute phase relies upon the availability of specific laboratory tests, which for DENV include molecular methods and nonstructural protein 1 (NS1) antigen detection. Anti-DENV IgM

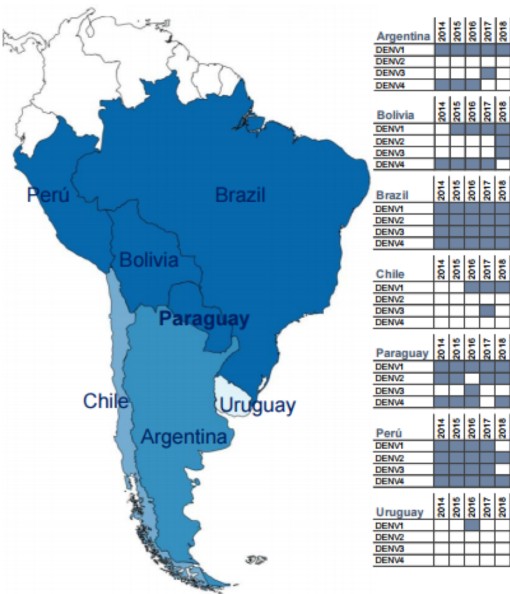

**Figure 1  Map of South America highlighting Paraguay and surrounding countries.** Inset tables show the dengue serotypes reported by each country to the Pan American Health Organization for the years 2014–2018 (data obtained from paho.org, accessed 16 August 2019). The included countries are shaded from dark to light blue according to the number of circulating DENV serotypes identified during this period (generated at https://mapchart.net/, CC BY-SA 4.0).

detection in acute-phase samples provides a presumptive diagnosis (*Peeling et al., 2010*; *World Health Organization, 2009*). As dengue can progress to severe disease, including plasma leakage, hemorrhage, and shock, ideal testing algorithms would not only detect DENV infections but also provide prognostic information.

The primary objective of the current study was to characterize DENV infections in the Central Department of Paraguay and the metropolitan area of Asunción. This region annually accounts for ∼2/3 of dengue cases in Paraguay and also reported Zika cases in 2016. Multiplex molecular testing, NS1 antigen detection and serological methods were implemented to confirm cases identified using a broad clinical case definition. We then sought to evaluate factors associated with dengue cases and the need for hospitalization in a predominantly adult patient population.

# MATERIALS & METHODS

## Ethics statement

The study protocol was reviewed and approved by the Scientific and Ethics Committee of the Instituto de Investigaciones en Ciencias de la Salud, Universidad Nacional de Asunción (IICS-UNA, IRB00011984), and the Emory University Institutional Review Board (IRB00000569). Written informed consent was obtained from all subjects. Children older than six years of age provided assent.

## Patient population and clinical samples

Patients of all ages were enrolled from January to May 2018 if they presented with an acute illness (≤8 days) defined by two or more of the following: fever (measured or subjective), red eyes, rash, joint pain involving more than one joint, and/or diffuse muscle pain. Patients with fever and no other localizing signs or symptoms were included. Day 1 was defined as the first day of symptoms. Exclusion criteria included dysuria or malodorous urine, cellulitis/skin abscess, vomiting and/or a productive cough. Patients were enrolled at in the Emergency Care Clinic at Hospital Villa Elisa and at IICS-UNA, both located in metro Asunción. The Emergency Care Clinic serves an ambulatory urgent care patient population; patients may be assigned to observation at Hospital Villa Elisa or referred to an inpatient facility that can provide a higher level of care. Serum was collected during the acute visit, aliquoted and stored at −80 °C until use. The results from hemograms, performed as part of routine care, were obtained by chart review. Data was included in this study if the hemogram was obtained on the day of the study visit ±1 day.

## Molecular detection

RNA was extracted from 140 μL of serum into 60 μL of elution buffer with the QIAamp Viral RNA Mini Kit (Qiagen, Germantown, MD, USA). All samples were tested for ZIKV, CHIKV and DENV by real-time RT-PCR (rRT-PCR) using a validated and published multiplex assay (the ZCD assay) as previously described (*Waggoner et al., 2016a*). DENV serotype and viral load were determined with a DENV multiplex assay using a published protocol (*Waggoner et al., 2013b*; *Waggoner et al., 2013c*). Samples that tested negative in the ZCD assay were tested for RNase P to confirm successful extraction and the absence of inhibitors (*Waggoner et al., 2013a*). All rRT-PCR testing was performed at IICS-UNA.

## Serological assays

All serum samples were test for DENV NS1 antigen and anti-DENV IgM and IgG using the STANDARD Q Dengue Duo assay (SD Biosensor, Suwon, South Korea). Results were read initially at 15 or up to 20 min, according to manufacturer recommendations. One hundred fifty-six samples were tested for anti-ZIKV IgG using the ZIKVG. CE kit (Diagnostic Bioprobes, Milan, Italy), which detects antibodies directed against the ZIKV NS1 antigen. Given a limited supply of anti-ZIKV IgG kits, a mixture of samples was selected for testing. This included dengue cases ($n = 76$) and non-dengue cases ($n = 80$), as well as include patients with anti-DENV IgG ($n = 58$) and without ($n = 98$). Assays were performed according to manufacturer recommendations.

## Definitions

Dengue cases were defined by either the detection of (1) DENV RNA in serum using the ZCD assay with confirmation in the DENV multiplex assay, or (2) both NS1 and anti-DENV IgM. This conservative definition was used to ensure the accuracy of dengue-case calls in the absence of paired acute and convalescent sera for confirmatory serological testing. This definition also allowed us to evaluate the performance of the STANDARD Q DENV NS1 assay, for which there was no prior published data. The sensitivity and specificity of
**Table 1 DENV diagnostic test results according to test method.** DENV viral load is shown for rRT-PCR positive samples within a given category.

| Test results | Composite definition | | Day of illness mean (sd) | Viral load mean (sd)[a] |
| --- | --- | --- | --- | --- |
| | Positive ($n = 119$) | Negative ($n = 112$) | | |
| Combination of methods | | | | |
|   rRT-PCR | 28 (23.5) | – | 3.1 (1.5) | 6.10 (1.69) |
|   rRT-PCR and NS1 | 60 (50.4) | – | 3.2 (1.4) | 7.85 (1.27) |
|   rRT-PCR, NS1, and IgM | 21 (17.7) | – | 5.3 (1.3) | 5.40 (1.19) |
|   rRT-PCR and IgM | 6 (5.0) | – | 6.7 (1.0) | 3.58 (0.22) |
|   NS1 and IgM | 4 (3.4) | – | 7.2 (0.1) | – |
|   Negative | – | 112 | 3.2 (1.6) | – |
| Positives according to method | | | | |
|   rRT-PCR | 115 (96.6) | – | 3.7 (1.7) | |
|   NS1 | 85 (71.4) | 4 (3.6)[b] | 3.9 (1.8) | |
|   IgM | 31 (26.1) | 7 (6.2)[b] | 5.6 (1.5) | |

**Notes.**
sd, standard deviation.
[a]Reported as $\log_{10}$ copies/mL of serum.
[b]Specificities were 96.4% (NS1) and 93.8% (IgM).

individual diagnostics were calculated in reference to positive and negative cases from this composite definition.

## Statistics

Basic statistical analyses were performed using Excel software (Microsoft, Redmond, WA). Univariate analyses and multiple linear regression analyses were performed using GraphPad Prism, version 8.0.1 (GraphPad, San Diego, CA, USA). Categorical variables were compared using Fisher's exact test. Age, day of illness, and continuous laboratory variables were compared by t test. Viral load comparisons were performed using non-parametric tests (Mann–Whitney with 2 groups; Kruskal-Wallis for 3 or more groups). Binary logistic regression analysis was performed using SPSS (IBM, Armonk, NY). Model fit was assessed by comparing $-2$ log likelihood statistics.

## RESULTS

Between January and May 2018, we enrolled 231 patients who met inclusion criteria, including 119 (51.5%) dengue cases and 112 (48.5%) non-dengue cases. No acute cases of ZIKV or CHIKV were detected. Of the dengue cases, 115 (96.6%) tested positive by rRT-PCR and 4 additional cases (3.4%) were positive for DENV NS1 and anti-DENV IgM (Table 1). All cases tested positive by rRT-PCR through day-of-illness 6 ($n = 104$), with rates of detection declining on days 7 (7/10, 70%) and 8 (4/5, 80%; Fig. 2). For the NS1 assay, the overall sensitivity and specificity were 71.4% and 96.4%, respectively (Table 1). Although there appeared to be an increase in NS1 sensitivity over the first 5 days of illness, this was not statistically significant ($p = 0.208$, day 5 vs. day 1–2; Fig. 2).
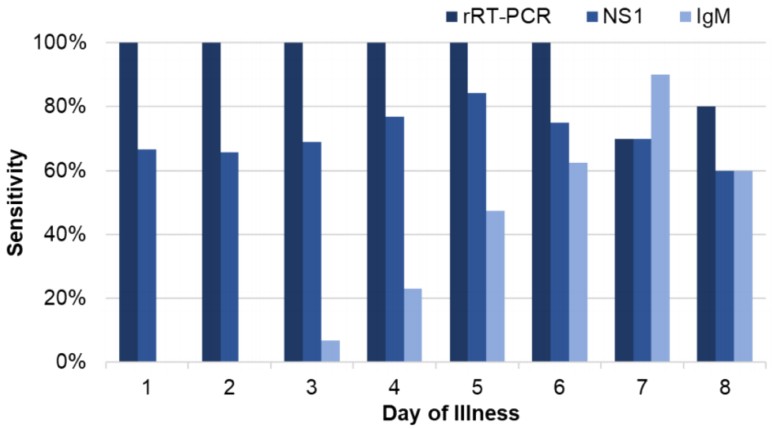

**Figure 2** Sensitivity of rRT-PCR, NS1, and IgM for dengue based on day of illness at presentation.

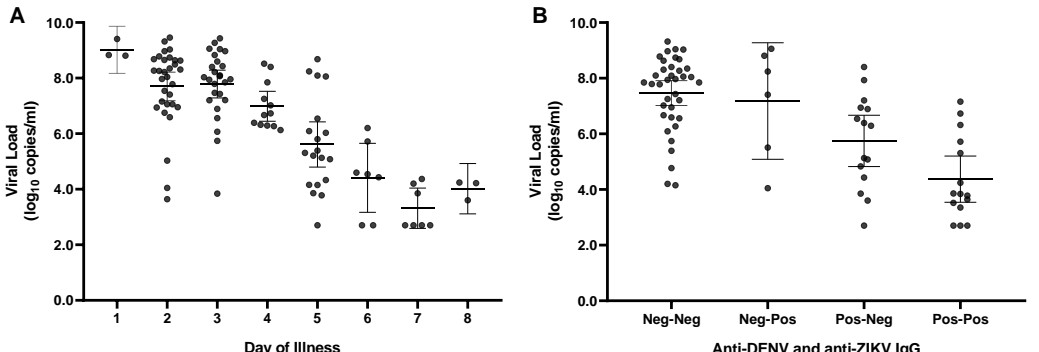

**Figure 3** DENV viral load at presentation based on day of illness and antibody status. DENV-1 viral load by day of illness at presentation (A). Viral loads are shown for individual samples; bars display the mean and 95% CI. Six patients had infections with DENV-2 ($n = 3$) or DENV-4 ($n = 3$), which are not displayed. DENV viral load at presentation decreases in a stepwise manner among individuals with anti-ZIKV IgG, anti-DENV IgG, or both (B). Results were significant by ANOVA for both analyses, $p < 0.0001$.

The overall sensitivity and specificity of anti-DENV IgM detection were 26.1% and 93.8%, respectively. The sensitivity of IgM detection increased from 0% on days 1–2 to 90% on day 7 ($p < 0.001$).

DENV serotype was determined in all 115 rRT-PCR-positive cases, with DENV-1 identified in 109/115 (94.8%) cases and 3 cases (2.6%) each of DENV-2 and DENV-4. No co-infections were detected. DENV-1 serum viral loads negatively correlated with day of illness at presentation (Fig. 3A), but too few data points were available for DENV-2 and -4 to draw meaningful conclusions. DENV viral load was also associated with NS1 detection: viral loads were significantly higher in samples with detectable NS1 (median 7.7 $\log_{10}$ copies/mL, IQR 5.8–8.6) compared to those in which NS1 was not detectable (median 5.6 $\log_{10}$ copies/mL, IQR 3.6–7.2; $p < 0.001$; Fig. S1).

Anti-DENV and anti-ZIKV IgG results were available for 156 patients, including 76 dengue cases (48.7%). 58 patients (37.2%) tested positive for anti-DENV IgG and 49 (31.4%) tested positive for anti-ZIKV IgG, with 32 patients (20.5%) positive for both. The viral load among dengue cases declined in a stepwise manner among patients with anti-ZIKV IgG, anti-DENV IgG, or both (Fig. 3B, $p < 0.001$ for the trend). In a multivariable model that included day of illness at presentation and patient age, DENV serum viral load was 1.3 $\log_{10}$ copies/mL lower among patients with detectable anti-DENV IgG compared to patients without anti-DENV IgG ($p < 0.001$, Table S1). Similarly, serum viral load was 0.7 $\log_{10}$ copies/mL lower among patients with anti-ZIKV IgG directed against NS1 ($p = 0.047$).

Among dengue cases, DENV NS1 detection was also associated with IgG status. Patients with anti-DENV IgG were significantly less likely to have detectable NS1 (20/36 (55.6%) vs. 65/83 (78.3%); OR 0.3, 95% CI [0.1–0.8]). However, when we controlled for the detection of IgG against ZIKV NS1, the OR for NS1 detection among dengue cases with anti-ZIKV IgG was 0.1 (95% CI [0.0–0.3]) and the association with anti-DENV IgG was no longer significant (OR 1.0, 95% CI [0.3–3.1]; Table S2).

## Epidemiologic characteristics

The epidemiologic characteristics of the patient population are shown in Table 2. This was predominantly an adult population, with only 11 participants <18 years of age at study entry (4.8%). Dengue cases occurred throughout the study period (Fig. S2) and were similar to non-dengue cases for the majority of epidemiological variables analyzed. Most patients reported living in a house or an apartment (157/188 for which data was available, 83.5%), but 31 patients reported "other" for housing without providing further detail. The odds of dengue in this population were significantly higher than among patients with a different living arrangement (OR 2.9, 95% CI [1.3–7.0]). Only 10.6% of our patients (21/199) reported having screens on their windows, though 79.2% of patients had air conditioning (156/197). The percentage of dengue cases among patients with neither screens nor air conditioning (19/37, 51.4%) was similar to that of patients with screens, air conditioning, or both (79/161, 49.1%; OR 1.1, 95% CI [0.5–2.2]).

A subset of patients self-reported having been vaccinated against yellow fever virus (YFV). The odds of having a dengue case were lower among patients who had received the YFV vaccine compared to those who had not (OR 0.6; 0.4–1.2), and more time had elapsed since vaccination among dengue cases. However, these trends did not reach statistical significance ($p = 0.15$). Receipt of the YFV vaccine did not increase the need for hospitalization among dengue cases.

## Clinical presentation

Patient symptoms at presentation are shown in Table 3. The majority of patients met inclusion criteria with fever plus one additional symptom in the study definition, most commonly muscle pain (198/225, 88.0%) and/or joint pain (172/221, 77.8%). Only 11 patients (4.8%) had fever and no other localizing sign or symptom (six dengue cases), and eight patients (3.5%) were enrolled that did not have fever (one dengue case). Patients who

**Table 2   Epidemiologic data on patients presenting with an acute febrile illness who tested positive or negative for DENV.**

| Factor[a] | Total | Dengue cases | Non-dengue | p-value |
|---|---|---|---|---|
| Patients | 231 (100) | 119 (100) | 112 (100) | |
| Gender, female | 128 (55.4) | 63 (52.9) | 65 (58.0) | |
| Age, mean (sd) | 31.94 (14.3) | 31.3 (15.0) | 32.6 (13.6) | |
| Clinical Site | | | | |
|     Hospital Villa Elisa | 185 (80.1) | 100 (84.0) | 85 (75.9) | |
|     IICS-UNA | 46 (19.9) | 19 (16.0) | 27 (24.1) | |
| Department | | | | |
|     Central | 209 (90.5) | 109 (91.6) | 100 (89.3) | |
|     Capital | 20 (8.7) | 8 (6.7) | 12 (10.7) | |
| Residence | | | | |
|     House | 149 (79.3) | 70 (74.4) | 79 (84.0) | |
|     Apartment | 8 (4.3) | 2 (2.1) | 6 (6.4) | |
|     Other | 31 (16.5) | 22 (23.4) | 9 (9.6) | 0.017 |
|     Screens | 21 (10.6) | 10 (10.0) | 11 (11.1) | |
|     Air-conditioning | 156 (79.2) | 76 (78.4) | 80 (80.0) | |
|     Running water | 199 (98.0) | 97 (97.0) | 102 (99.0) | |
|     Water storage | 18 (8.8) | 8 (8.0) | 10 (9.5) | |
| Exposures | | | | |
|     Travel in the last month | 52 (25.0) | 22 (20.8) | 30 (29.4) | |
|     Work or school outside of the home | 152 (80.4) | 68 (73.9) | 84 (86.6) | |
|     Work or school outdoors | 24 (55.8) | 11 (57.9) | 13 (54.2) | |
| Medical History | | | | |
|     Received yellow fever vaccine | 71 (42.0) | 31 (37.3) | 40 (46.5) | |
|         Years since vaccination, mean (sd)[c] | 7.6 (4.1) | 8.5 (3.7) | 6.8 (4.3) | |
|     Personal history of dengue | 78 (34.2) | 29 (24.6) | 49 (44.6)[d] | 0.002 |

**Notes.**

n, number; sd, standard deviation.

[a] Unless otherwise specific, all values presented as n (% of patients with a response recorded).

[b] OR for dengue for patients reporting "other", 2.9 (95% CI [1.3–7.0]).

[c] Year of YF vaccination was available for 52 patients (24 DENV-positive, 28 DENV-negative).

[d] OR for dengue in patients who reported a history of dengue, 0.4 (95% CI [0.2–0.7]).

reported red eyes were significantly more likely to have dengue (OR 2.1; 95% CI [1.2–3.6]) and those with a sore throat were significantly less likely to have dengue (OR 0.5; 95% CI [0.3–0.8]; Table 3). Although a reported headache increased the odds of having dengue, this did not reach statistical significance (OR 2.3; 95% CI [1.0–5.5]), and headache was very common overall. Other symptoms occurred with similar frequency in the two groups, and no combination of symptoms accurately differentiated between dengue and non-dengue cases.

Hemogram results are also shown in Table 3. Patients with dengue had significantly lower platelet and leucocyte counts relative to non-dengue cases (Fig. 4). Thrombocytopenia (<150,000 per µL) and leucopenia (<4,000 cells/mm$^3$) were both significantly associated with DENV infections (Table 3). However, patients with both findings were not at greater odds of having a DENV infection (OR 8.9; 95% CI [3.4–23.0]) than patients

**Table 3  Symptoms and laboratory findings among patients with and without dengue.**

| Factor[a] | Total | Dengue cases | Non-dengue | OR (95% CI)[b] | p-value |
|---|---|---|---|---|---|
| Patients | 231 (100) | 119 (100) | 112 (100) | | |
| Day of symptoms, mean (sd) | 3.9 (2.5) | 4.1 (1.9) | 3.7 (3.0) | | |
| *Symptoms and signs at presentation* | | | | | |
| Fever | 221 (96.5) | 117 (99.2) | 104 (93.7) | | |
| Headache | 206 (89.6) | 111 (93.3) | 95 (85.6) | 2.3 (1.0–5.5) | 0.083 |
| Retro-orbital pain | 94 (40.9) | 53 (44.5) | 41 (36.9) | | |
| Muscle pain | 198 (88.0) | 99 (86.8) | 99 (89.2) | | |
| Joint pain | 172 (77.8) | 92 (80.7) | 80 (74.8) | | |
| Nausea | 142 (61.7) | 73 (61.3) | 69 (62.2) | | |
| Malaise | 119 (51.7) | 62 (52.1) | 57 (51.4) | | |
| Red eyes | 99 (45.0) | 61 (51.5) | 38 (35.8) | 2.1 (1.2–3.6) | 0.010 |
| Abdominal pain | 95 (41.3) | 51 (42.9) | 44 (39.6) | | |
| Vomiting | 73 (31.7) | 41 (34.5) | 32 (28.8) | | |
| Diarrhea | 66 (28.7) | 31 (26.1) | 35 (31.5) | | |
| Shortness of breath | 64 (27.8) | 36 (30.3) | 28 (25.2) | | |
| Sore throat | 61 (26.5) | 23 (19.3) | 38 (34.2) | 0.5 (0.3–0.8) | 0.011 |
| Cough | 51 (22.2) | 25 (21.0) | 26 (23.4) | | |
| Rash | 52 (23.1) | 32 (27.8) | 20 (18.2) | | |
| Edema | 37 (16.2) | 17 (14.3) | 20 (18.0) | | |
| Bleeding | 32 (13.9) | 20 (16.8) | 12 (10.8) | | |
| *Laboratory results* | | | | | |
| Hemoglobin, g/dL, mean (sd) | 13.9 (1.5) | 14.0 (1.5) | 13.8 (1.5) | | |
| Platelet count, per $\mu$L, mean (sd) | 217,550 (89,921) | 188,227 (82,079) | 252,609 (86,650) | | <0.001 |
| Thrombocytopenia, <150,000 per $\mu$L | 46 (22.8) | 36 (32.7) | 10 (10.9) | 4.0 (1.9–8.2) | <0.001 |
| Leucocyte count, cells per mm$^3$, mean (sd) | 6090 (3686) | 4158 (2023) | 8401 (3899) | | <0.001 |
| Leucopenia, <4,000 cells per mm$^3$ | 73 (36.1) | 63 (57.3) | 10 (10.9) | 11.0 (5.1–22.2) | <0.001 |

**Notes.**

CI, confidence interval; OR, odds ratio; sd, standard deviation.

[a]Values presented as *n* (%) unless otherwise indicated, percentages were calculated based on the number of patients with data recorded for a particular variable.

[b]OR of having a dengue case versus a non-dengue case.

with leucopenia alone (OR 11.0, 95% CI [5.1–22.2]). Dengue cases had lower neutrophil and lymphocyte counts, but these occurred in proportion to the decrease in leucocyte counts (see Supplemental Files, Raw Data).

## Hospitalization

For the analysis of factors associated with hospitalization for dengue, we focused on cases that presented to Hospital Villa Elisa, as only 1/19 dengue cases (5.3%) at IICS-UNA required hospitalization. Of 100 dengue cases at Hospital Villa Elisa, 26 (26.0%) were hospitalized and one patient died (Table 4). A number of clinical and laboratory findings were associated with hospitalization in univariate analysis. Rash and bleeding were more common among hospitalized cases. Admitted patients were significantly more likely to have detectable anti-DENV IgG and IgG against both DENV and ZIKV (anti-NS1). Despite the presence of anti-DENV IgG, viral load was significantly higher among admitted patients,

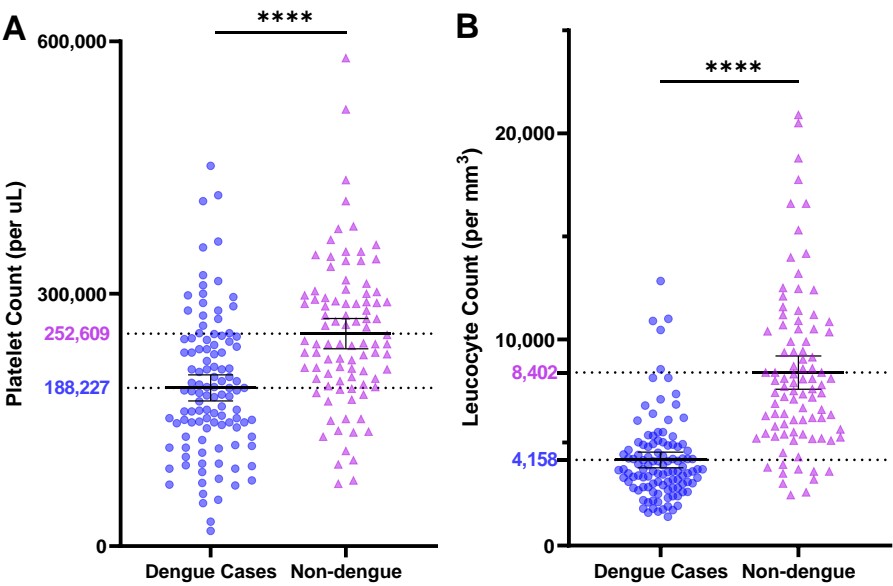

**Figure 4 Platelet (A) and leucocyte (B) counts at presentation among dengue cases (●) and non-dengue cases (▲).** Bars represent means ±95% CI; population mean values are shown.

but there was no difference in NS1 detection. In multivariate analysis, the best-fit model for predictors of hospitalization only included platelet count and day of illness, though the odds ratio for day of illness did not reach significance (OR 1.3, 95% CI [0.9–1.8]; Table S3).

## DISCUSSION

In the current study, we characterized a set of dengue cases in a primarily adult population that presented to outpatient facilities in metro Asunción. Dengue is a major public health problem in Paraguay, with adults accounting for a significant proportion of cases. At Hospital Villa Elisa, 58% of patients with an acute febrile illness were adults ≥20 years of age, and an additional 13% of patients were aged 15-19. While studies have demonstrated that the clinical presentation of dengue in adults may differ from that in children (*Hammond et al., 2005*; *Kittigul et al., 2007*; *Low et al., 2011*; *Potts & Rothman, 2008*), less research has specifically evaluated factors that differentiate dengue from other causes of an acute febrile illness in the adult population (*Chadwick et al., 2006*; *Gregory et al., 2010*; *Low et al., 2011*). All but one dengue case in our study presented with fever and a high percentage of cases had headache, myalgia and/or arthralgia (*Chadwick et al., 2006*; *Hammond et al., 2005*; *Kittigul et al., 2007*; *Potts & Rothman, 2008*). These symptoms are consistent with the previous reports of dengue in adults, but were common among both dengue cases and non-dengue cases (*Low et al., 2011*). The only two symptoms that were significantly associated with dengue in our population were red eyes and the absence of a sore throat. Red eyes have not been commonly associated with dengue (*Chadwick et al., 2006*), though one prior study found an association with DENV-1 (*Yung et al., 2015*). The absence of a sore throat has

**Table 4  Clinical history and test results among hospitalized and outpatient dengue cases at Hospital Villa Elisa.**

| Patient factors[a] | Total | Hospitalized | Outpatient | OR (95% CI)[b] | *p*-value |
|---|---|---|---|---|---|
| Patients | 100 (100) | 26 (100) | 74 (100) | | |
| *History and Clinical findings* | | | | | |
| Gender, female, n (%) | 52 (52.0) | 12 (46.2) | 40 (54.1) | | |
| Age, mean (sd) | 31.6 (14.5) | 36.5 (20.0) | 29.9 (11.6) | | 0.044 |
| Day of illness | 3.81 (1.84) | 5.0 (2.4) | 3.4 (1.4) | | <0.001 |
| YFV vaccination | 23/64 (35.9) | 5/16 (31.2) | 17/48 (35.4) | | |
| Past dengue, per report | 34/99 (34.3) | 12/25 (48.0) | 22/74 (28.6) | 2.2 (0.9–5.5) | 0.143 |
| Rash | 28/96 (29.2) | 13/25 (52.0) | 15/71 (21.1) | 4.0 (1.5–10.0) | 0.005 |
| Diarrhea | 27/100 (27.0) | 11/26 (42.3) | 16/74 (21.6) | 2.7 (1.0–6.9) | 0.070 |
| Bleeding | 18/100 (18.0) | 10/26 (38.5) | 8/74 (10.8) | 5.2 (1.8–14.1) | 0.006 |
| *Dengue test results* | | | | | |
| rRT-PCR, positive | 99 (99.0) | 25 (96.2) | 74 (100) | | |
| Viral load, mean (sd) | 6.44 (2.04) | 6.76 (1.84) | 5.51 (2.35) | | 0.028 |
| NS1 | 69 (69.0) | 17 (65.4) | 52 (78.4) | 0.8 (0.3–2.0) | 0.632 |
| IgM, anti-DENV | 25 (25.0) | 10 (38.5) | 15 (20.3) | 2.5 (1.0–6.6) | 0.112 |
| IgG, anti-DENV | 28 (28.0) | 14 (53.9) | 14 (18.9) | 5.0 (1.9–12.2) | 0.002 |
| IgG, anti-ZIKV | 19/70 (27.1) | 7/16 (43.8) | 12/54 (22.2) | 2.7 (0.9–8.1) | 0.114 |
| IgG against both DENV and ZIKV | 13/67 (19.4) | 7/13 (53.8) | 6/54 (11.1) | 9.3 (2.2–36.3) | 0.002 |
| *Laboratory results[c]* | | | | | |
| Hemoglobin, g/dL, mean (sd) | 14.1 (1.4) | 14.0 (2.0) | 14.2 (1.2) | | |
| Platelet count, per µL, mean (sd) | 191,563 (85,951) | 119,250 (77,402) | 215,667 (74,749) | | <0.001 |
| Thrombocytopenia, <150,000 per µL | 31 (32.3) | 18 (75.0) | 13 (18.1) | 13.6 (4.5–43.2) | <0.001 |
| Leucocyte count, cells per mm$^3$, mean (sd) | 4167 (2135) | 4814 (3209) | 3952 (1604) | | 0.087 |
| Leucopenia, <4,000 cells per mm$^3$ | 55 (57.3) | 13 (54.2) | 42 (58.3) | | |

**Notes.**

CI, confidence interval; OR, odds ratio; sd, standard deviation.

[a]Values presented as n (%) unless otherwise indicated.

[b]OR for hospitalization versus outpatient care.

[c]Lab results were available for 24 and 72 hospitalized cases and outpatients, respectively.

been associated with dengue in a previous series (*Gregory et al., 2010*). However, this was only reported by 26.5% of our patients overall, which limits the utility of this finding in clinical practice.

In contrast to clinical findings, the results of general laboratory studies differed significantly between dengue cases and non-cases. Leucopenia and thrombocytopenia were associated with dengue (ORs 11.0 and 4.0, respectively), a finding that has been consistently documented in previous studies (*Biswas et al., 2012*; *Kalayanarooj et al., 1997*; *Low et al., 2011*). However, patients with both findings did not have higher odds of dengue that those with leucopenia alone, which may have resulted from temporal differences in the development and resolution of these abnormalities (*Biswas et al., 2012*). The nadir leucocyte counts occurred on days 5–6 after symptom onset, whereas platelet counts demonstrated a consistent decline through day 8 (see Supplemental Files, Raw Data). Many factors were significantly associated with hospitalization in univariate analyses but were also strongly correlated with one another (viral load, antibody status, platelet count,
day of illness). Given the sample size, our ability to model all of these factors in logistic regression was limited, and admission decisions were likely based on the platelet count, which may have obscured the association between other factors and disease severity.

DENV infections were confirmed using a combination of methods, though all but four cases were positive by rRT-PCR (115/119, 96.6%). NS1 was detected in 71.4% of infections and proved specific for DENV (96.4%). Notably, the performance of this commercial NS1 kit has not been published, but results appeared similar to those reported for other rapid NS1 assays (*Blacksell et al., 2011*). Consistent with previous observations, viral loads were significantly higher among NS1-positive individuals (*Duong et al., 2011*; *Duyen et al., 2011*; *Erra et al., 2013*; *Tricou et al., 2011*). Both viral load and NS1 detection were significantly associated with the detection of anti-DENV IgG and anti-ZIKV IgG, which in this study was directed against the NS1 antigen. In an earlier study, ZIKV-specific neutralizing antibodies were not detected among a subset of our patients (A Rojas, 2019, unpublished data). As such, anti-ZIKV IgG identified by ELISA in the current study is favored to represent cross-reacting anti-DENV antibodies. In the subset of patients with results for both IgG assays, the presence of anti-ZIKV NS1 IgG accounted for virtually all false-negative NS1 results. Although such antibodies have been known to reduce NS1 detection in secondary cases (*Jayathilaka et al., 2018*; *Lee et al., 2015*; *Lima Mda et al., 2014*), the pathophysiologic significance of anti-NS1 antibodies in human DENV infections remains unclear (*Glasner et al., 2018*; *Jayathilaka et al., 2018*). We demonstrate that these antibodies can be detected in the acute-phase and, in combination with anti-DENV IgG, are more common among hospitalized dengue cases. These serologic findings combined with an elevated DENV viral load warrant further evaluation using standardized severity criteria (*World Health Organization, 1997*; *World Health Organization, 2009*).

Dengue cases were less likely to report living in a house or apartment (recorded as "other" in the study questionnaire). This was also observed in a seroprevalence study in Mexico where these patients reported a "shared" living arrangement (*Pavia-Ruz et al., 2018*). Other aspects of the home environment evaluated in our study did not differ between dengue and non-dengue cases. The absence of air conditioning and window screens did not appear to increase the risk for DENV infection. However, complete screening of the home and air conditioning have been associated with decreased vector indices and dengue incidence in other settings (*Manrique-Saide et al., 2015*; *Pavia-Ruz et al., 2018*; *Reiter et al., 2003*; *Waterman et al., 1985*), and the addition of screens has been proposed as a means of DENV control through improvements to the built environment (*Lindsay et al., 2017*; *Vazquez-Prokopec, Lenhart & Manrique-Saide, 2016*). Our findings may indicate that patients acquired DENV outside the home or that the use of these interventions is incomplete (e.g., non-intact screens, intermittent use of air conditioning). Determining the location of exposure will have important implications for DENV control efforts in metro Asunción.

Vaccination against YFV is not part of the routine schedule in Paraguay, and as a result, our patient population included a mixture of individuals who did or did not report receiving the vaccine. There was no evidence of increased risk from YFV vaccination for either incident dengue or the development of severe disease. These data are consistent with

recent findings from Brazil where no association was found between severe dengue and receipt of the YFV vaccine (*Luppe et al., 2019*).

DENV-1 was the predominant serotype identified in the current study. This is consistent with recent DENV epidemiology in Paraguay but precluded a comparison of symptoms caused by each serotype. DENV-1 is less commonly associated with severity than DENV-2, though severe and debilitating illness still occurs (*Balmaseda et al., 2006*; *Low et al., 2011*; *Thomas et al., 2014*). Clinical findings in our patients appear more consistent with dengue in adults rather than dengue caused specifically by DENV-1, which is often associated with lower rates of arthralgia and myalgia (*Burattini et al., 2016*; *Martins Vdo et al., 2014*; *Suppiah et al., 2018*; *Yung et al., 2015*). An additional limitation to the study is that we were unable to evaluate the performance of the clinical case definition for different arboviral infections, and in particular ZIKV infections that may not present with fever (*Braga et al., 2017*). Finally, patients were included who reported up to 8 days of symptoms prior to enrollment. Laboratory data from day 8 produced conflicting results and raises questions regarding the accuracy of symptom recall past one week. These data support the use of earlier enrollment cut-offs with scheduled follow-up visits to monitor the kinetics of certain laboratory findings.

## CONCLUSIONS

In this study, we sought to characterize DENV infections in a predominantly adult population in Paraguay, focusing on the region with the highest dengue incidence, metro Asunción. This work highlighted clinical, epidemiologic, and laboratory factors that are associated with DENV detection in the acute setting and the potential role of specific antibodies in diagnosis and the progression of disease. Future directions will involve the prospective evaluation of how factors identified in the current study associate with and may predict dengue severity.

## ACKNOWLEDGEMENTS

We thank the members of the study team based at the Instituto de Investigaciones en Ciencias de la Salud, Universidad Nacional de Asunción, and Hospital Villa Elisa in Paraguay for their dedication and excellent work, and we are grateful to the study participants and their families. The authors would like to thank Diagnostic Bioprobes who kindly provided the ZIKVG.CE kits used in this study as well as Muktha Natrajan and Varun Phadke for their thoughtful comments during the preparation of this manuscript.

### Funding

Research was supported by National Institutes of Health grant K08 AI110528 (Jesse Waggoner). In addition, the development of this collaboration was supported by funding from the Consejo Nacional de Ciencia y Tecnología (CONACYT) in Paraguay (Alejandra

Rojas: PVCT16-66 and Jesse Waggoner: PVCT17-65). The funders had no role in study design, data collection and analysis, decision to publish, or preparation of the manuscript.

## Grant Disclosures
The following grant information was disclosed by the authors:
National Institutes of Health: K08 AI110528.
Consejo Nacional de Ciencia y Tecnología (CONACYT): PVCT16-66, PVCT17-65.

## Competing Interests
The authors declare there are no competing interests.

## Author Contributions
- Alejandra Rojas and Jesse Waggoner conceived and designed the experiments, performed the experiments, analyzed the data, contributed reagents/materials/analysis tools, prepared figures and/or tables, authored or reviewed drafts of the paper, approved the final draft.
- Fátima Cardozo performed the experiments, analyzed the data, contributed reagents/materials/analysis tools, prepared figures and/or tables, authored or reviewed drafts of the paper, approved the final draft.
- César Cantero, Sanny López and Cynthia Bernal performed the experiments, analyzed the data, prepared figures and/or tables, authored or reviewed drafts of the paper, approved the final draft.
- Victoria Stittleburg performed the experiments, analyzed the data, authored or reviewed drafts of the paper, approved the final draft.
- Francisco Eugenio Gimenez Acosta analyzed the data, authored or reviewed drafts of the paper, approved the final draft.
- Laura Mendoza, Benjamin A. Pinsky, Ivalena Arévalo de Guillén and Malvina Páez analyzed the data, contributed reagents/materials/analysis tools, prepared figures and/or tables, authored or reviewed drafts of the paper, approved the final draft.

## Human Ethics
The following information was supplied relating to ethical approvals (i.e., approving body and any reference numbers):

The study protocol was reviewed and approved by the Scientific and Ethics Committee of the Instituto de Investigaciones en Ciencias de la Salud, Universidad Nacional de Asunción (IICS-UNA, IRB00011984), and the Emory University Institutional Review Board (IRB00000569).

## Data Availability
The raw data are available in the Supplemental Files. All data has been de-identified to protect study participants.

## Supplemental Information

Supplemental information for this article can be found online at http://dx.doi.org/10.7717/peerj.7852#supplemental-information.

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
