# Peer review of "Characterization of dengue cases among patients with an acute illness, Central Department, Paraguay"

_PeerJ, doi:10.7717/peerj.7852_

## Round 0.1 · original submission · Minor Revisions

You are advised to address the comments of the reviewers and carefully revise the manuscript, indicating the exact changes made in the manuscript via tracked changes.

Reviewer 1 ·

Basic reporting

The manuscript is well written with appropriate context and table & figure. The authors have provided underlying data in very clear and properly organized format for any interested reader to re-analyze or integrate into another study.

Experimental design

The research questions are well defined and experiments and data collection are sound. The methods are described or cited from literature.

Validity of the findings

The data presented in this manuscript adds important information to the Dengue epidemiology knowledge. The authors have done a great job in organizing the data, applying statistics and presenting all examined parameters including the non-significant ones.

Additional comments

Though this manuscript is about Dengue in Paraguay, it would be informative to add in the introduction the presence and serotypes of dengue in and around the geographical site and study time. In the preceding years the Africa-Asia region had predominantly Dengue 2. A map in this context would be very helpful.
Similarly, Dengue from other places have been reported more or less in the same format and parameters. Please add in the discussion how the symptoms associated with Dengue 1 found in this study match or differ compared to other Dengue serotypes and patient populations.
Figure 3: Please reduce the number of asterisks representing statistical significance. Its clear that the traditional * < 0.05, ** < 0.01, *** < 0.001 does not necessarily add significance to the scientific comparison and depends on other statistical parameters like n. In the current set the data points from the two groups are mostly overlapping, and it would be really difficult to put a cut-off limit if they were really distinct.

Reviewer 2 ·

Basic reporting

Clear and unambiguous, professional English used throughout. YES

Literature references, sufficient field background/context provided. YES

Professional article structure, figures, tables. Raw data shared. YES

Self-contained with relevant results to hypotheses. NO

Experimental design

Original primary research within Aims and Scope of the journal. YES (to Paraguay)

Research question well defined, relevant & meaningful. It is stated how research fills an identified knowledge gap. NO

Rigorous investigation performed to a high technical & ethical standard. YES

Methods described with sufficient detail & information to replicate. YES

Validity of the findings

Impact and novelty not assessed. Negative/inconclusive results accepted. Meaningful replication encouraged where rationale & benefit to literature is clearly stated. YES

All underlying data have been provided; they are robust, statistically sound, & controlled. YES

Conclusions are well stated, linked to original research question & limited to supporting results. NO

Speculation is welcome, but should be identified as such. YES

Additional comments

This article is very important to Dengue studies in Paraguay, main because describe the last epidemic perfil in country in adults. But, I have some questions, listed below.

1. About number of patients. The authors think that with 231 patients (119 Dengue cases positives) could suggest just title viral and anti-DENV IgG positive as severity markers? Other factors could be included, such as comorbidity or pre-existing disease.

2. Many of the results are expected, aren't novelty. For example, little viral load to more days disease, after 6 day principally. I suggest having this information along with the results already expected and demonstrated in the literature, emphasizing that results corroborate what is already recognized.


Other suggestions to methods sections:

- Patient population and clinical samples.

Line 103 - 107: "Patients of all ages were enrolled from January to May 2018 if they presented with an acute illness (≤ 8 days) defined by 2 or more of the following: fever (measured or subjective), red eyes, rash, joint pain involving more than one joint, and/or diffuse muscle pain. Patients with fever and no other localizing signs or symptoms were included.

Reviewer: Could write as follow: "Patients of all ages were enrolled from January to May 2018 if they presented with an acute illness (≤ 8 days) defined by fever (measured or sujective) only or with 1 or more of the following: red eyes, rash, joint pain involving more than one joint, and/or diffuse muscle pain.

- Molecular detection.

Line 124: "Arboviral rRT-PCR testing was performed at IICS-UNA."

Reviewer: "All rRT-PCR testing was performed at IICS-UNA" (If this information is right, if not, rewriter).

---

## Round 0.2 · accepted · Accept

Thanks for submitting your revised manuscript. It is a pleasure to accept your manuscript in its current form for publication.

Reviewer 1 ·

Basic reporting

no comment

Experimental design

no comment

Validity of the findings

no comment

Additional comments

This is a very well presented and informative manuscript.

Reviewer 2 ·

Basic reporting

The paper have clear and unambiguous, professional English used throughout, literature references, sufficient field background/context provided, professional article structure, figures, tables, raw data shared and self-contained with relevant results to hypotheses.

Experimental design

no comment

Validity of the findings

no comment

Additional comments

The authors made some the requested modifications and justified the refusal of the others suggested.